# Towards a Model of Urban Evolution—Part I: Context

**Daniel Silver** [1] , **Patrick Adler** [2] **and Mark S. Fox** [1,*]

1   School of Cities, University of Toronto, Toronto, ON M1C1A4, Canada
2   Department of Geography, University of Hong Kong, Polfulam, Hong Kong
*   Correspondence: msf@eil.utoronto.ca

**Abstract:** This paper seeks to develop the core concepts of a model of urban evolution. It proceeds in four major sections. First, we review prior adumbrations of an evolutionary model in urban theory, noting their potential and their limitations. Second, we turn to the general sociocultural evolution literature to draw inspiration for a fresh and more complete application of evolutionary theory to the study of urban life. Third, building upon this background, we outline the main elements of our proposed model, with special attention to elaborating the value of its key conceptual innovation, the "formeme". Last, we conclude with a discussion of what types of research commitments the overall approach does or does not imply, and point toward the more formal elaboration of the model that we undertake in "Towards a Model of Urban Evolution, Part II" and "Towards a Model of Urban Evolution, Part III". In "Towards a Model of Urban Evolution, Part IV" we demonstrate the application of the model to Yelp data.

**Keywords:** urban evolution; urban modelling; urban signatures

## 1. Introduction

The urbanization [1] of the world has received increasing attention over recent decades. A common refrain is that cities have now become the major form of human settlement. Many discussions about urbanization start by noting the percentages of people who now inhabit cites as opposed to more dispersed human settlement types. Correlatively, there has been a recent upsurge of interest in urban issues across a number of academic fields, including Geography, Economics, Sociology, Political Science, Planning, Design, Engineering, Computer Science, and more.

While each field understandably tackles urban questions from their own disciplinary point of view, there are a number of basic themes that cut across them. Some of these include: the emergence of urban order from local interactions [2–5], the diffusion of innovations [6,7], the sources of widely shared, possibly universal patterns of urban scaling and form on the one hand [8,9], and locally distinctive variants on the other [10–12], the relative contribution and value of bottom-up market and sorting processes vs. top-down planning interventions [13], sources of persistent patterns of segregation and integration [14,15], processes by which cities both unleash and block human capacities for creativity, community, growth, and prosperity [16,17], the mutual interplay between locations and the networks and relations within which they are embedded [18–20], the mechanisms by which cities maintain recurrent activity and settlement patterns while also exhibiting periodic bursts of sudden transformation [21,22], and more. In its most general form, the ambition to pull these types of themes together into an overarching framework amounts to the pursuit of a "new science of cities" [23]. In broad terms, the goal has been to develop more sophisticated descriptions and classifications of the urban environment; to model or simulate urban interactions in a way that can facilitate a better historical understanding of urban development; and to make or test predictions about how urban areas have developed or might develop into the future.

There is much intellectual ferment around developing the concepts and categories that would constitute such a science. Multiple possibilities are on the table, which we will review in more detail below. However, no consensus has emerged, and the field is ripe for consideration of a wide variety of options. In this paper, we propose that a renewed attention to evolutionary theory can provide a keystone for building a new science of cities, extending concepts derived from biological and ecological science. An evolutionary approach, we argue, promises to provide a model for explaining the nature and distribution of urban characteristics, how they come into being, and why the number and functions of their constituent members change. In other words, akin to evolutionary explanations of other sociocultural domains such as religion [24], it promises a unified explanatory model of urban life.

Making good on this theoretical proposition requires first developing a more formal model to describe the phenomena that are critical to an urban evolutionary approach, which in turn can structure efforts to understand urban evolution in simulations and to formally apply evolutionary concepts to urban data. The primary goal of this paper is to move toward a theory of urban evolution by articulating the main elements of such a model. The model and the rich expressive possibilities it opens up are not themselves a "theory"—in the sense of a set of explanatory propositions. Instead, the goal is to develop and illustrate a framework in which many theories about urban evolution may be formulated.

An evolutionary approach to socio-cultural phenomena is not new, and the effort stretches back at least to the mid-19th century, if not further [25]. Yet across a number of fields, researchers have recently sought to expand, synthesize, and apply core evolutionary ideas to social and cultural phenomena, often abandoning "stage" theories of developmental, linear progress for a more complex adaptation of Darwinian concepts. For example, evolutionary economists, evolutionary economic geographers, and organizational ecologists—in collaboration with biological theorists—speak of "generalized Darwinism" [26]: a high-level algorithm, consisting of variation, differential reproduction, and retention, as applicable to organizations and firms as it is to amoebas and foxes [27,28]. Historians and anthropologists adapt phylogenetic methods to uncover lineages embedded in historical texts and artifacts [29–31], while cognitive anthropologists of religion adopt tools from population genetics to discern features of religious ideas that make them more or less likely to become embedded in and recalled by human minds [32]. These contribute to a more general anthropological effort to develop a science of cultural evolution that holds insights into the "secret of our success" [33], whereby cultural innovation and transmission enhance and interact with human biology and intelligence. Students of technology study how tools of daily life such as forks, hammers, or paper clips have evolved into their present form through a series of intermediate variants [34,35]. Furthermore, some sociologists even attempt macro-historical studies of how human societies—individually and as a system—have evolved [36–38]. These efforts dovetail with renewed interest in applications of evolutionary thinking to social research in general and in sociocultural evolution in particular, with for example both *The Oxford Handbook of Evolution, Biology, and Society* and Turner and Machalek's *The New Evolutionary Sociology* appearing in 2018 and the creation of the Cultural Evolution Society in 2016.

Neither is the application of evolutionary concepts to cities altogether new. Urban researchers have often noted similarities between cities and evolutionary processes and phenomena. Concepts such as ecology, niches, self-organization, scaling, selection, and DNA loom large [39–48]. These arise because cities exhibit numerous phenomena with clear evolutionary dimensions: they show recurrent combinations of continuity and change; they reproduce and borrow elements from one another, while at the same time constituting distinct environments [49] that induce local translations and adaptations; they compete with one another and are themselves hosts to ongoing competitions for space and resources; they exhibit enduring and durable patterns of order that emerge from the interactions and plans of numerous individuals and groups, in ways that go beyond the designs of any of them separately; and more.

The interest in evolutionary concepts is rooted not only in the promise of making sense out of characteristic urban phenomena. Evolutionary theory also offers a powerful way to answer basic "why" questions of urban life for which other explanatory approaches seem ill-suited, such as those rooted in design (i.e., planning) or necessity (i.e., environmental pressures). For example, if we ask why some styles of built form (such as porches or cul-de-sacs) or settlement patterns (such as segregation by occupation or ethnicity) predominate, appealing to the goals and plans of individual or collective actors will provide a partial account at best. The question is which of various options become selected and retained—questions of evolutionary selection mechanisms. Similarly, if we ask why various physical or social features of some cities are more or less similar to one another, common environment pressures provide only a partial picture. The question is also one of inheritance and lineage, of the diffusion and gradual alteration of characteristics across contexts that generate shared, derived characteristics from common origins, and of how contingent starting points constrain future possibilities along a directed path—questions of evolutionary taxonomy and the mechanics of "descent with modification". The appeal of urban evolutionary theory, like that of evolutionary theory in general, arises from its promise of a powerful way of asking and answering "why" that deviates sharply from traditional alternatives [50,51].

Yet urbanists have tended to pursue these insights piecemeal, without seeking to join them into a general evolutionary model, and to employ evolutionary concepts in a somewhat loose and metaphorical way, in terms of general dynamics of change and development. Pumain's observation in 1998 remains apt: "There has been very little . . . serious research for introducing the concepts of evolution in urban theory, despite a more and more extensive use of the word itself" [52]. There are signs of change. For example, Dibble et al. [53] take some initial steps toward viewing cities as "evolved cultural products" characterized by "cumulative adaptation" and "successive accumulation of technologies and complexities". However, they acknowledge that this program is in its infancy, as do evolutionary economic geographers who anyway tend to focus on the evolution of organizations and technologies in an urban environment more than how cities and the institutions surrounding and supporting organizations and firms themselves evolve [41,54,55]. While urban scholars recognize the potential of evolutionary theory, a general model is still lacking to guide and synthesize the effort.

This series of papers is an effort to begin to think through from the ground up a conceptual model of urban evolution. We seek to articulate a formal model that includes basic constituents of the urban evolutionary process, and provides an expressive language that can be applied and extended further. Though many ideas embedded in our model are not new, we have sought to synthesize elements often featured by other urbanists, who have variously stressed physical infrastructure [21], sociocultural dimensions [43], scale [56], diffusion [57], activity spaces [58], emergence [13], and path dependence [59], among other related concepts. We do not seek to determine a priori all relevant variables and how they should be stratified. Yet the model does aim to provide a vocabulary for representing, evaluating and comparing potential variables, and guiding questions about how these variables might fit together.

This synthesis of various strands of urban theories with sociocultural theories of evolution is the primary contribution of this paper. In particular, we aim to develop a framework for conceptualizing the mechanisms and processes involved in the genesis, continuity, spread, transformation, and integration of the elements that combine to form cities. We feature the various, often interconnected, physical, social, and cultural features that comprise cities, such as road patterns, architectural features, organizational forms, ethno-cultural neighbourhoods, industrial and occupational concentrations, amenity mixes, as well as numerous opportunities for interaction, chance encounters with strangers and acquaintances, broad social networks, and diverse forms of activities. In other words, traits of urban life are our units, and our model provides a framework for formulating propositions about how various forms of urban life appear at different rates in different

places and for classifying such forms in terms of their shared or divergent trajectories. In this view, a given city is at once a complex set of solutions to the problems of urban existence and the medium in which new problems unfold. A formal model of urban evolution promises to provide a new and powerful set of tools by which we can understand where the characteristics of our cities come from, and the mechanisms through which they have survived, spread, and been carried forward through time.

We develop these general insights in three parts. The first asks what a model of urban evolution is, and pursues this question through a critical review of prior efforts moving in this direction. It also articulates the entity that constitutes the core of our model, the *formeme.* This concept, we argue, helps to advance urban thought, as it lays the basis for the very possibility of a theory of urban evolution. Building on this groundwork, Part II develops the formal tools for modeling urban evolution, defining its key terms and functions. Part III uses these tools to build an evolutionary model in terms of (1) sources of variation; (2) principles of selection; and (3) retention.

Part I proceeds in four major sections. First, we review prior adumbrations of an evolutionary model in urban theory, noting their potential and their limitations. Second, we turn to the general sociocultural evolution literature to draw inspiration for a fresh and more complete application of evolutionary theory to the study of urban life. Third, building upon this background, we outline the main elements of our proposed model, with special attention to elaborating the value of its key conceptual innovation, the "formeme". Last, we conclude with a discussion of what types of research commitments the overall approach does or does not imply. This paper series is part of the Urban Genome Project, about which more information can be found here: https://academic.daniels.utoronto.ca/urbangenome/ (accessed on 28 March 2022).

## 2. What Is a Model of Urban Evolution?

As in social sciences more generally, evolutionary thinking in urban studies is not new. However, the main forms it has taken have tended to be either partial or misleading. Many classifications of these efforts are possible and illuminating. For instance, Mehmood [60] divides appropriations of evolutionary concepts into planning theory between "vitalist", "organicist", and "natural evolutionary" metaphors. The goal of this section's critical review and synthesis however is less taxonomic and more focused on situating what we take to be the main aims and contributions of our approach in reference to other efforts, with the broader ambition of elaborating a model of urban evolution inspired by Darwinian concepts as they have been developed in the general socio-cultural evolution literature.

## 3. Steps toward Evolutionary Thinking in Urban Theory

This section undertakes a critical review of prior and ongoing efforts that more or less explicitly use evolutionary concepts to study urban life. In particular, we discuss the application to cities of stage models, ecological models, complex adaptive systems ideas, DNA and path dependence models, and scaling models. The overarching argument is that, while these offer a rich and valuable set of ideas, they also tend to conflate developmental processes with evolutionary processes, fail to build their theories around the core evolutionary "algorithm" of variation, selection, and retention, or feature one aspect of the evolutionary process in isolation from the others. Because of this, a fully evolutionary model has yet to be developed in the urban sciences.

Our review spans multiple disciplines—not from a spirit of eclecticism but in the conviction that doing so is a necessary pre-condition for building a general model, that no single discipline could encompass such a model, but that many fields have crucial insights to offer. This will invariably involve oversights and perhaps simplifications that might be unsatisfying to specialists. Yet every once in a while, we may wish to rise up from our particular domains to meet at a higher altitude, and see what we can learn from there.

### 3.1. Stage Theories

Perhaps the most common form of urban theory termed "evolutionary" is a variant of "stage theories" of society. Stage theories of cities and neighborhoods posit some (purportedly) necessary sequence of stages through which they must pass. The history of urban studies is replete with such theories: Hoover and Vernon's [61] five stages of the "neighborhood life cycle" (development, transition, downgrading, thinning out, renewal); Geddes' [62] three stages of urban development (primary, secondary, tertiary); Mumford's six stages [63] (Eopolis, Polis, Metropolis, Megalopolis, Tyranopolis, Necropolis); Birch's [64] six stages (Rural, First Wave, Fully Developed/High Quality/Residential, Packing, Thinning, Recapture); Taylor's [65] four stages (Infantile, Juvenile, Mature, Senile).

Like their even grander cousins from the likes of Spencer, Comte, and Parsons that posit stage theories of societies as a whole, urban stage theories, while embodying some potentially important insights, are not properly evolutionary theories, in the Darwinian sense [66]. Rather, they are developmental theories, most similar to theories of the steps by which an organism grows from immaturity to maturity to senility—a fact most evident in Taylor's scheme, but an orientation common to the others as well. Regarding Geddes, Marshall and Batty [67] gives a somewhat fuller picture: "Overall, Geddes' evolutionary urbanism was therefore part 'developmental' (city-as-organism), part 'evolutionary' (in a non-Darwinian way) and part 'environmental' (city as environment, rather than organism)". Marshall and Batty trace Geddes' influence on urban planning through his students, especially Patrick Abercrombie, and then systems theorists, and on to Batty's own attempt to reinvigorate "physicalist" approaches that seek to solve social problems by manipulating the built environment. This is not to deny that many cities and neighborhoods do exhibit sequences approximating those envisioned by these theories. Yet this sort of emergence of new forms of subsistence (primary, secondary, tertiary) and social organization (Metropolis, Megalopolis, etc.) is more similar to what biological evolutionists call "grades" or "major transitions" [51] (p. 5). Though developmental transitions of this kind are important, they are only one part of evolutionary logic, which involves "innovation and recombination, differential proliferation under selection, and diversification . . . which characterizes the branching tree of an evolutionary process" [51] (p. 4).

Yet not only are stage theories more developmental than evolutionary; in their historical forms they have often rested upon dubious or even pernicious assumptions. The most damning of these is an implicit teleology: that movement through these stages is necessary and inevitable. Notably, this assumption has been responsible for urban variants of eugenics [66], for instance in justifying disinvestment from neighborhoods presumed to be on their way toward the end of their lives, and to hasten them toward their "inevitable" demise [68]. Though this kind of teleological thinking is anathema to Darwinian evolutionary thought, its association with evolutionism in the social sciences has made many social scientists understandably wary. A contemporary framework for studying urban evolution must therefore be built on different grounds.

### 3.2. Chicago School Ecology

A different approach traces its origins to the Chicago School of Sociology, which pioneered "ecological" styles of thought about cities, and social life in general—though it built on predecessors such as von Thünen [69,70]. More explicitly drawing on biological inspirations, this type of account views the city as a competition for space that generates interdependent zones, characterized by their proximity to the central business district. Chicago School sociologists moreover sought to develop dynamic accounts of how the nature and occupants of these zones would change, for example through groups pushed or pulled by ecological forces to "invade" one zone or the other, setting off conflicts that could result in various outcomes from displacement to novel forms of integration.

Here too some critical components "f an'evolutionary theory are pointed toward, but only in a partial way. For example, while competition for survival is crucial to any evolutionary account of why some forms of life persist or disappear, the ecological dimension of

the Chicago School tended to exclusively feature "economistic" notions of competition [71]. Yet even in the biological domain, selection proceeds through many mechanisms, including sexual selection based on aesthetics and taste [72]—an aspect of urban life that the Chicago School did pursue, but under a different heading ("community area" mapping of "ways of life") that it never integrated with its ecological ideas [73]. Moreover, the Chicago School focused more or less exclusively on competition and intra-urban change, but had little to say about how urban innovations in one place spread elsewhere or produce longer-term lineages defined by shared, derived characteristics. This is ironic in that their own maps became iconic models that influenced the spread of urban forms in their image [74]. Even if some of these same problems remain, nevertheless Hawley's work [75] along with others in the (now somewhat marginalized) human ecology tradition continued to develop the early Chicago School ecological insights to a high degree of theoretical sophistication, and provide a rich fund of ideas from which a contemporary general model of urban evolution can draw.

### 3.3. Complex Adaptive Systems

The image of cities as an ecology of emergent interdependent zones has proven inspirational to a more recent wave of research that treats cities as "complex adaptive systems", often invoking "evolution" as part of this program. A key background to much of this work has been an effort to come to grips with some of the challenges faced by urban planning since the second half of the twentieth century. Portugali's narrative [13] is an instructive window into this version of recent urban intellectual history (see also Batty and Marshall [67]). On this account, enthusiasm for top-down planning ran against the "first planning dilemma", namely that efforts to scientifically predict and technically control the urban process tended to be either empty or ineffective [76]. Many urban thinkers turned instead to phenomenological, humanistic, and Marxist approaches [13]. These brought a humane sensitivity to deep organizing principles of cities (often under the heading of "modes of production") and the possibility of their radical transformation. Yet they too had difficulties providing guidance in the here and now. The result was postmodern pessimism to the effect that urban life was too fragmented to be understood, let alone directed or cultivated in any meaningful way.

As a way out of this conundrum, and animated by constructive critics of top-down planning like Jane Jacobs and Christopher Alexander, "complexity" and "self-organization" have become central concepts to many urban researchers. Elsewhere, philosophers like Edgar Morin have discussed complexity as a core epistemological concept and a successor to rationalistic modes of thinking. Our framework is more applicable to cited authors who are interested in dynamic models of social systems such as cities. Comparisons to evolutionary concepts have been a part of this program.

"Here too, life in general and urbanism in particular, are like one big theatre in which we the audience-scientists-planners sit and look and often respond to the dynamics of this changing complexity, and by so doing we—like the audience in the theatre—also play our role in this ongoing 'game of life'" [13].

Batty makes the Darwinian connection explicit, referring to his "new science of cities" [67] as "neo-Darwinian" and linking this to the conviction that cities are dynamic, open systems, forever "far-from-equilibrium", exhibiting long-term but nevertheless temporary "order parameters" [13], while being susceptible to both gradual change and abrupt transition [67].

Yet perhaps because its primary intellectual inspiration has been physics, though this work has brought the term "evolution" and certain evolutionistic ideas back to the fore, it has not developed them to their fullest potential. For example, Batty and colleagues' work invokes evolutionary themes in models of how cities' coordinated and ordered patterns emerge from a multitude of bottom-up decisions that "adapt, mutate and innovate with respect to an individual's action space" [76]. This insight is developed primarily through cellular automata models that show the emergence of global patterns from out of simple

more localized interactions, as Xie and Batty make explicit with their "integrated urban evolutionary modeling" suite [77]. Yet while these models are powerful illustrations of emergence and self-organization, at their base are sets of spawning rules by which cells aggregate to form complex patterns—a process more akin to embryology than evolution [51]: "Multicellular development and evolution are deceptively similar. Both involve growth (the proliferation of cells in development, of individuals in evolution), a branching process (differentiation in development, diversification including speciation in evolution), and interactions among the branches (normally cooperation based on a division of labour in development but any or all of competition, conflict and cooperation in evolution)" [78–80]. Still, in developing an explicitly "physicalist" approach to these phenomena, this work provides valuable insights about the physical side of urban evolution, which can be extended into a synthetic model that seeks to join the physical more explicitly with the social, economic, and cultural.

It is this side that has been the focus of Juval Portugali's work [13]. Indeed, in one of his computer "city games" that simulate the "self-organization" of the city, he seeks to introduce Darwinian themes somewhat more directly into models in which (physicist) Hakken's "synergetics" is nevertheless the theoretical core. In a combined cellular automata/agent-based model, he endows agents with a "cultural program" that he likens to a "meme". These memes define agents' cultural identities, and as the agents interact (in search of housing) they give rise to groups which either foster the replication of that identity (through creating collectivities to support them) or lead them to disappear. Here we do have the seeds of a genuinely evolutionary model, but one which primarily highlights how the urban environment shapes the evolution of cultural attributes. While a crucial step, a full urban evolutionary theory of the city needs to also consider the evolution of the urban environment itself—that is, it needs to synthesize Portugali's evolving agents with an equally evolutionary theory of Batty's physical forms.

### 3.4. Urban DNA and Path Dependence

Other researchers have turned to evolutionary concepts in somewhat different ways. Some refer to "urban DNA," [47,79,80] or "architectural genotypes" [48], and again often in connection with similar themes of complexity and adaptation [79–83]. Though this is often largely an undeveloped metaphorical connection, there are some underlying themes with real evolutionary resonance, in particular by pointing toward taxonomic methods and the generation of path dependencies by which existing conditions are retained into the future. For some, identifying "urban DNA" involves a classification effort to gather key indicators and group cities into related clusters based on shared attributes [84]. For others, urban and architectural genotypes refer to the underlying principles revealed in the surface appearance of space, such as the "space syntax" [78] lying behind the seemingly illegible tangle of intersections constituting an urban grid.

Wilson [85,86] and Delmelle [11] represent variants of a different approach that features path dependence more prominently. Wilson for example seeks to build mathematical models out of structural variables such as population characteristics, housing, economic indicators, transit availability, and the like: "The structural variables can then be considered to represent the 'DNA'—the 'genetics'—and we can explore whether the elements of these can be combined into 'genes' as explanatory variables for different forms of evolution [85]. These structural variables provide "initial conditions" that lay out "paths of development" [86], which, if changed, can produce abrupt "phase transitions". Taking a more data-driven approach, Delmelle uses sequence analysis to identify characteristic, relatively long-term trajectories of neighborhoods across dozens of U.S. cities, allowing classifications of cities not only in terms of shared attributes but also in terms of similar historical pathways [39]. From a somewhat different theoretical tradition (historical institutionalism), Sorensen [87] codifies and synthesizes main elements of path-dependent models relevant for the history of urban planning, highlighting processes such as positive feedback loops (where initial conditions make their repetition more likely), critical junctures

(contingent confluences of events that set the parameters for the future), and endogenous changes (incremental changes produces by the processes internal to the paths themselves, such as the inertia of municipal boundary location over time producing governmental fragmentation in growing regions).

Taken as a whole, these lines of research show promise in highlighting the importance of initial and inherited conditions in constraining and even propelling the types of variations that can appear in cities. In terms of the basic Darwinian process of variation, selection, and retention, they outline directions for describing mechanisms and outcomes of retention by which cities "remember" and transmit their current conditions in similar future iterations, thereby generating the characteristic evolutionary pattern of "descent with modification". Moreover, they begin to move in the direction of systematic evolutionary taxonomies of cities based on modern computational classification techniques.

At the same time, as recent contributors acknowledge, these lines of research fall short of a truly phylogenetic taxonomy in terms of lineages [53]. They rarely confront the significant challenges that adapting biological notions and methods of phylogenetics and taxonomy would require. For example, given the myriad forces of regulation and standardization in cities (such as industrial processes of manufacturing, top-down urban planning, etc.) it is an open question whether, and to what extent, phylogenetic conclusions can ever be drawn from morphological analysis. Understanding the interplay and relative balance of horizontal and vertical transmission is a core challenge in similar attempts to import phylogenetic concepts into archaeological investigation of artifacts, a problem which becomes more acute in efforts to more rigorously adapt biological methods of phylogenetic tree construction [88].

These methodological questions touch on thorny conceptual challenges about the extent to which distinctions such as genotype vs. phenotype neatly apply in the domain of urban and cultural evolution. Sometimes also discussed under the heading of "replicator" vs. "interactor" [89], this is a much-debated question in the literature on socio-cultural evolution that goes beyond the scope of the present discussion. Put schematically, however, our view is that the topic is best approached in functional and evolutionary terms: functionally, we may consider "genotype" to refer to heritable information and "phenotype" to non-heritable traits, which may change through an entity's existence without being passed on [90]. Evolutionarily, the extent to which a given urban form exhibits a strong or weak distinction between replicating and interacting dimensions is itself a result of evolutionary processes that exhibit wide variation. For example, a building located in a district with a strict building code more closely approximates the biological distinction between genotype vs. phenotype: changes made to the building during its life course will rarely be passed on, unless they become written into the local planning code. In less formally planned settings, the range of heritable information is much wider: nearly any perceptible feature of the building can be copied and transmitted, with more or less fidelity. Thus, rather than define a priori the line between genotype and phenotype, we prefer to develop a language for formulating propositions about where and why that line may vary. The concepts of "recoding costs" and "signals" are crucial to this component of our model. More generally, what is needed is an effort to elaborate the general Darwinian concept of inheritance or retention of form in a way suited to the domain-specific problems of urban evolution, and to combine a model of retention with models according to which novel urban variants emerge and spread.

*3.5. Scale*

A final stream, also inspired by themes of complexity and self-organization, has brought evolutionary themes to urban studies via the topic of scale and in the process highlighted some themes neglected in the other traditions reviewed above. One line of interest highlights the fact that larger cities tend to produce innovations that in turn diffuse throughout the urban system. These innovations eventually become incorporated and retained as a new set of standard conventions [91], before the cycle of innovation–diffusion–

routinization is repeated anew. Another is more directly rooted in biology, and features "allometric scaling laws" [8,9,92]. Like their biological cousins, such laws are grounded in purportedly basic constraints that cities face in distributing energy and resources to all their parts. These constraints in turn produce certain regularities across cities of all sizes. For instance, the number of gas stations or grocery stores in a metropolitan area can be predicted to a very high degree of accuracy based upon its relative size alone, as can crime or patent rates [92]. Crucial in this literature is the observation that cities exhibit not only economies of scale (where infrastructure can serve more people) but also "super-linear" scaling, a phenomenon apparently unknown in the biological domain. This means that some features (such as crime and innovation) appear at higher rates in bigger cities, which also implies to some observers that as cities grow they face compounding challenge to innovate at increasing rates: a doubling of population requires more than double the level of innovation [92]. Taken together, these findings have indicated to some evolutionists (e.g., Ridley [93]) that cities, like organic bodies, evolve in regular and predictable ways owing to the common pressures under which they exist, regardless of their local particularities.

As in biology [51], urban scaling laws have proven controversial. One major issue concerns the definition of city boundaries. Depending on the definition (central city, municipal, county, metro area, region, and more) the type and direction of the "universal laws" varies dramatically [94,95]. While it is possible to argue for one definition as better than another, in their efforts to identify the spatial level at which "universal" scaling unfolds, previous studies have left relatively unexamined evolutionary mechanisms that generate and diffuse hierarchically embedded urban forms of different degrees and sizes in the first place, and the way the resultant hierarchies create environments—i.e., construct niches—that affect the evolution of urban life within them. What are the conditions of buildings becoming embedded in blocks, blocks in neighborhoods, neighborhoods in cities, cities in regions, and so on? What forms of urban life thrive (or not) when embedded in different environments? In this way, by extending concepts of ecological niche construction further into urban studies (already a major topic in the study of organizations, cf. Hannan and Freeman [96]), we can build upon and carry further the insights of West and others and seek to make scale itself something to be explained on evolutionary grounds rather than being exclusively explanatory [94,97].

In sum, recent transdisciplinary urban research has exhibited renewed interest and hope for building a "new science of cities" with some key evolutionary scaffolding. While certain themes have been central—most notably stages, ecological competition, path dependence, emergence, and self-organization—a striking feature of these efforts has been the relative absence of an integrated model based on what are arguably the central mechanisms of Darwinian evolution: variation, selection, and retention. As a result, these efforts have been relatively silent on the problem of explaining the nature and distribution of urban characteristics, how they come into being, and why the number and functions of their constituent members change. Such questions are the core of a theory of urban evolution. Similarly, some recent work in architecture and urban design moves in this direction, most notably Scheer's [98] examination of the recurrence of a few formal typologies across space and the challenges inherent in purposeful efforts to work against the endogenous evolutionary processes that give rise to them. Such work provides a fund of examples and descriptive insights about how the urban evolutionary process operates on the ground, without formulating the tools for a more general explanatory model. To move in this direction, we now turn to the literature on sociocultural evolution.

### 4. From General Darwinian Sociocultural Evolution to Urban Darwinian Evolution

Having suggested that the diverse fields of urban studies lack a unified model of urban evolution, we may find inspiration for articulating such a model from other fields. In this section we turn to the literature on general Darwinian sociocultural evolution and seek to draw out core principles to apply and adapt to the urban domain. This extension of recent transdisciplinary interest in evolutionary thinking to urban life is a relatively unexplored

subject. Indeed, as Mehmood [60] notes: "planning theory has largely avoided any direct use of Lamark's and Darwin's key evolutionary concepts in urban planning with a few exceptions ... despite the fact that evolutionary strands have increasingly become stand-alone fields of inquiry, as apparent in the case of evolutionary anthropology, economics, sociology, and cultural evolution" [60]. Hence, our goal is to elaborate in broad terms key concepts from the "evolutionary turn" in the social sciences, and then to reflect on what it could mean to translate those insights into the terms of urban theory. We highlight four key insights: social learning; the evolutionary approach as an application of a high-level "algorithm" of variation–selection–retention to a specific domain; the nature of sociocultural evolutionary units; and capacity of evolutionary theories to synthesize existing theories of sociocultural change.

First, models of sociocultural evolution tend to feature social learning processes. To be sure, there are at least two other major approaches to the general question of sociocultural evolution, in addition to the "social learning or meme-based" approach: "the gene-based biological (sociobiology, human behavior ecology, and evolutionary psychology) ... and dual inheritance or gene-culture coevolutionary theory" [51]. However, the vast majority of social scientific applications of evolutionary theory are of the second type. In this approach, biological processes and elements (i.e., genes) do not feature centrally. Rather, the accent is on how various socio-cultural entities are adopted and passed on. There is certainly room for gene-based or gene-culture coevolutionary studies of cities, and as new large-scale, geocoded datasets of genetic data appear, the possibility becomes more tractable (for some recent examples in this direction, see [30,99,100]). Nevertheless, given that the social learning approach is more well-developed, it is reasonable to build a model of urban evolution upon it, and potentially incorporate other approaches in future iterations. Hence our model embodies a sociocultural approach that makes urban characteristics its central units, where, as we develop further below, "urban characteristics" is understood in a broad way that includes not only cities' physical forms but also the forms of groups and activities they support.

Second, sociocultural approaches develop explanatory models around a tightly linked set of a few core concepts: variation, selection, and retention [28,101]; though the specific terms sometimes differ e.g., [42,54,102–104]. "Variation" refers to novel cultural forms, whether idealities (such as memes, concepts, mental plans), artifacts and their properties [34], or something in between (e.g., "habits and routines" [26]). Novel sociocultural variants appear through many processes, from intentional planning to trial-and-error experimentation, luck, error, and conflict. As variations accumulate, differences between or within entities become more sharply delineated, and some replicate more widely than others, typically through the transmission of ideas, models, memes, knowledge, information, beliefs, and values [66]. The mechanisms by which some proliferate while others do not constitute "selection", which generally depends on an entity's capacity to utilize available resources in the environment [105]. "Retention" refers to systems of inheritance or memory, by which past successful forms constrain and direct future options, such as through schools, legal codes, common beliefs, as well as the material artifacts of society (written records, machinery, and the built environment) [26]. In some forms, the variation–selection–retention triad can overemphasize survival at the expense of transmissbility. As especially Part II and III make clear in our discussion of signals, our modelign approach includes a robust place for transmissibility.

By repeated cycles of variation–selection–retention, sociocultural entities become increasingly adapted to their environments (organizational, cognitive, geographical) and vice versa. This process gives rise to emergent orders with the appearance of design yet without any single purposeful, deliberate designer, as well as to gradual changes through time that eventually branch into distinct lineages. This is a dynamic order. Existing conditions tend to positively feedback into future iterations; incremental change is ongoing yet sudden disruptions are always possible. Accordingly, our model elaborates its core concepts in terms of the appearance of novel urban variants, their differential probabilities

of being selected, and of being retained into the future. We explain the differential forms and functions of cities by modeling them in terms of this Darwinian algorithm.

Third, whatever the term, in sociocultural approaches to evolution, the central unit is generally not concrete biological individuals. Rather, it is some set of "iss and oughts" [51]—some notion of what things are and what they are for—encoded in roles, identities, or artifacts. The specific terms vary widely by discipline and author, from the sociologist's norms and values to the anthropologist's traditions and mores to the linguist's rules and competencies to the economist's conventions, habits, and routines to Dennett's "information worth copying", and beyond. These culturally "loaded" entities become more or less prevalent. For instance, roles such as "molecular biologist" out-compete cell biologists [51] or the competitive advantage of certain firms at one time leads to enduring competitive advantage for the regions containing those firms because successful routines are better replicated locally [42]. Richards [106], following Campbell [27], goes so far as to apply an evolutionary model to the theory of evolution itself, in which multiple sets of evolutionary ideas co-exist and compete within the selective pressures of a broader intellectual social environment. He highlights the explanatory advantages of the evolutionary approach over other models of the history of science, such as revolutionary models, Kuhn's gestalt model, Popper's falsification model, or Lakatos' research program model.

Given that biological individuals are generally not the unit of sociocultural evolution, the analytical challenge is to fundamentally re-orient our perspective from the standard anthropocentric view. This perspectival shift is akin to asking what makes one gene or another more likely to survive in a given "vehicle"—that is, the organism in which it is lodged, or even its "extended phenotype" [107] that includes for instance not only the beaver's teeth but its dam. In the sociocultural domain, this means seeing the world from the point of view of ideas or artifacts (or the information encoded therein) and asking about what "strategies" would make them more or less likely to survive and be copied and reproduced. The answer will generally revolve around differential success at attracting human interest. For example, Boyer [24] "explains religion" by seeing the world from the point of view of religious ideas and asking what features of such ideas make them more memorable and sharable to human minds. The recurrent set of religious themes we find in the world are highly evolved responses to this problem, "experts" at getting themselves remembered and passed on. In this way, identifying the core evolutionary unit for a particular domain, and adopting its point of view, is perhaps the most fundamental step toward leveraging the explanatory power of evolutionary thinking.

A model of sociocultural urban evolution therefore involves a reversal of perspective. It sees the world from the point of view of features and models of urban life—the BID, the cul-de-sac, the Bohemian Neighbourhood, the porch—and investigates what "strategies" lead human agents to be more or less likely to select and retain them. As in sociocultural evolution in general, it is the interaction between the two—human beings, with both relatively stable characteristics as well as diverse tastes, resources, interests, and the like; sociocultural entities with varying attributes—that explains the existing array and proportions of alternatives in a population, and their trajectories. In the case of sociocultural evolution, the populations in question are populations of sociocultural entities. Similarly, in a model of sociocultural urban evolution, the populations in questions are populations of urban features that may spread, be copied, and make their (always slightly altered) recurrence in the future more or less likely.

Fourth, theorists of sociocultural evolution point toward the synthetic power of evolutionary theories. Consider two examples, one from organizational theory and the other from sociological theory. Aldrich, Ruef, and Lippmann [108] compare evolutionary theories of organizational change to ecological, institutional, interpretative, organizational learning, resource dependence, and transaction costs approaches. They demonstrate that each approach features some dimension of change, such as the focus in ecological theories on population level changes in organizational foundings and disbandings, or in institutional theories on the adoption of organizational norms and customs. Yet, as they demonstrate,

evolutionary theory has a synthetic power to flexibly integrate special insights of each approach into a general theory of change.

Similarly, Koch, Silvestro, and Foster [109] argue that recent sociological theories of cultural change have tended to rely on notions of purposeful activity (such as the activism of social movements) or exogenous shocks (such as major events that delegitimize accepted conventions) while paying relatively little attention to endogenous changes across populations of actors. They contend that an evolutionary theory of culture provides the conditions for a general sociological theory of cultural change, by specifying key conditions any theory must meet, such as: a material basis (e.g., "cultural forms" like musical genres); mechanisms for the replacement, emergence, and persistence of such forms ("mortality", "innovation", and "stability"); as well as their transmission ("learning"). They add metatheoretical principles to this list, such as the condition that mechanisms must be intelligible both from the side of actors and from the side of the cultural forms and that a theory must be able to make predictions about whether stability or change is more likely in various settings. While Koch, Silvestro, and Foster note that various existing sociological theories of change are consistent with this model, only an evolutionary model meets all these criteria for a general theory of social change.

A model of urban evolution holds a similar synthetic promise of a general theory of urban change. As in [109], it encompasses insights from other perspectives on urban change, such as ecological or diffusion models, while meeting the general criteria for a successful theory of change. For example, our model defines the material basis of urban change not in cultural forms but in urban forms ("the formeme"), elaborates mechanisms for the emergence, selection, retention, and transmission of variants, and provides analytical tools for examining these processes as an outcome of interactions between urban forms, and the expectations they have of their users, and human users, and the expectations they have of urban forms. Moreover, in Part III we outline conditions under which populations of forms are expected to change or persist. In these ways our model aims to extend this synthetic power of evolutionary theories of culture into the domain of urban research.

## 5. Preliminary Outline of a Model of Urban Evolution

To be sure, disagreements persist among sociocultural evolutionists (cf. [25]). Some are about terminology. Others are more philosophical, about the best way to understand the relationship between sociocultural and biological evolution—for instance, whether one is analogous to the other, or both are instances of a more general universal process to be elaborated in particular ways for specific domains (e.g., [27,50,108,110,111]). A recurrent question concerns whether sociocultural evolution is Darwinian or Lamarckian. As Blute [51] (p. 10) notes, many researchers are initially drawn to the proposition that sociocultural evolution is Lamarckian in that it seems to operate through purposive learning of better (i.e., more adaptive) practices that are then passed on through directed learning. Yet the Darwinian point is not that people do not seek to create and disseminate valuable cultural innovations, it is that there is no clear connection between a given innovation and its proliferation. Most new products go nowhere; and as any student of urban history knows, must innovations in urban design do not catch on. Those that do often do so in ways and places their creators did not imagine. And many innovations that spread, while perhaps being initially designed to be beneficial to their inventors, prove to be quite disastrous—a case in point is AIG's experience with mortgage-backed securities. 'There is no evidence in any area of human endeavour that, as a statistical body, innovations are biased in the direction that would be required for them to spread successfully. In fact, most fail. Sociocultural evolution, like the biological, is Darwinian rather than Lamarckian" Blute [51] (p. 18). Yet for present purposes we can set these aside and move to the substantive question of the central features and challenges of a theory of urban evolution. Here we overview the main features of the model that are developed more fully in Parts II and III before discussing some of the theoretical advantages of the model.

### 5.1. Main Features of Model

The model we develop in detail in Parts II and III is an attempt to elaborate these features and meet these challenges in precise and rigorous terms, building up from basic definitions to more elaborate propositions via a logically consistent formal set-theoretical language. Nevertheless, it is useful to lay out the main lines of the approach in a preliminary and primarily verbal way. Following the sociocultural evolution literature reviewed above, the most basic issue is to articulate the unit of urban evolution. Having done that, we can examine how new variants of these units appear, strategies by which some become more or less likely to survive in different environments, and processes by which they become retained in future iterations of the urban landscape and thereby constrain and direct the unfolding of evolutionary lineages. Hence the focus of this section is on elaborating the main unit of our model—the formeme—and discussing what it means to see the world from the "form's eye" point of view. In aspiration, our approach is similar to that of Storper and Scott [112] in seeking general processes that hold across time and space. This is in keeping with the promise of Darwinian theory in general, to bring the diversity of natural evolution under a common model. Even so, such an approach requires close examination of local and special mechanisms by which the generic process of variation-selection-retention occurs. Nevertheless, in contrast to urban theorists like Scott and Storper, we are not developing a theory of urbanization per se but rather elaborate the terms that make formulating hypotheses about urbanization and other aspects of urban evolution possible. For this reason in the present work we are agnostic regarding the definition of concepts like "city", since we view that too as a result of evolutionary processes to be investigated [98].

Taking our cue from the sociocultural evolution literature, we start from the proposition that the units of urban evolution are themselves sociocultural entities, such as buildings, roads, parks, neighborhood types, porches, building codes, city plans, zoning regulations, and the like. While this might seem like a "physicalist" orientation, we also follow the sociocultural evolution literature in proposing that these entities embody "iss and oughts". They are evolved responses to problems about how to organize space, not only in terms of physical design, but also in terms of who and what the space is for, that is, in terms of its characteristic groups and activities [98]. Therefore, the basic units of urban evolution should encode information about how a space is physically organized, and who and what it is for.

Building on this insight, our model proposes the concept of the formeme as the basic unit of urban evolution. A formeme is a specific encoding of urban space as a combination of physical features and the groups and activities toward which they are oriented. A formeme is a way of physically organizing space for some sets of activities and groups. This definition might also be summarized as a script or set of instructions. "Be made out of this stuff arranged in this way, for doing these things for groups of people like this". Accordingly, we may define the complete information about how a space is physically and socially organized as its urban genome.

Seen in this way, the cities we observe are large populations of such scripts. They are a collection of survivors, exhibiting strategies that have rendered them increasingly likely to be recurrently adopted by human agents within particular sets of local environments, or what amounts to the same thing, increasingly difficult for them to abandon. As in evolutionary theory more generally, this does not imply that they are globally optimal according to an overarching universal standard, or good in any moral sense, but only that they have achieved a degree of relative fitness to the local setting that may in fact be detrimental to their success in other times or places. In fact, a question we examine in Part III concerns hypotheses about general vs. specific strategies and the types of forms that come to embody them.

A central task of the formal model we develop in Parts II and III is to rigorously elaborate the intuitions behind the concepts of formeme and urban genome, and their role in the Darwinian algorithm of variation–selection–retention. The value of such formalization comes in part in revealing the logical skeleton of informal models, but even

more in identifying their key assumptions and omissions, as well as necessary auxiliaries and correctives. In the ideal case, formalization will make evident new connections and implications, and produce an "idealized" model against which empirical phenomena can be measured.

To formalize a model of urban evolution, we develop a set-theoretical representation in Part II. In a general sense, our approach to modelling urban evolution follows Richerson and Boyd's [113] (p. 2) characterization of cultural evolution where "sets of traits are transmitted by a given society". In our case the traits, which we call elements, span forms, groups and activities and are distinguished by time and place. We are not attempting to make quantitative predictions but represent elements that are duplicated, modified and transmitted over time and space. Hence, the use of sets is appropriate. Our model makes it possible to formally state propositions of variation, selection, and retention that can be tested using urban data. In this representation, a formeme is composed of members from three sets: P, G, and A. P is the set of all possible physical forms; G is the set of possible users (or groups); A is the set of possible uses (or activities). A formeme is a particular combination of members of these sets, which our model represents by rendering them as indices of f: f[p], f[g], f[a]a. An "urban genome" (U) is a set of formemes f tied to a particular location and time: its physical organization and the uses and users toward which it is more geared.

This basic representation is supplemented with a number of other terms, most crucially what we call H and S. H refers to the human uses and users of a space and allows us to model the potential gap between their expectations and those toward which a space is relatively oriented. We use the same representational scheme to define their expectations for a space in terms of P, G, and A: h[p], h[g], h[a]. If U is what the space expects from its users, H is what the people expect from their spaces. S refers to signalling processes by which formetic information is communicated and transmitted to other locations and potential users. We define a number of properties of S, such as their communication method, reach, precision, audience, clarity, and noise. Since features of S affect the scope and fidelity of formeme transmission, they strongly shape their evolution. Together, U, H, and S constitute what we call the signature of a given space at a given time, which provides a complete representation of the features relevant to our evolutionary model.

In addition to these basic terms, we develop a number of functions by which such terms can be utilized in constructing evolutionary models. These include similarity, evolutionary trajectories, formeme survival, and activity costs and recoding. The core "similarity" function we develop, *fdist*, returns the distance between the elements of two formemes. We develop several variations of *fdist* in Part II, but elaborate in particular how one can be used to identify evolutionary trajectories of urban genomes with properties such as divergence, convergence, volatility, and pace. Additional functions capture the existence of paths and lineages, through functions we call *Gpath* (any path between genomes) and *Dpath* (directed paths). To codify conditions of formeme survival, we define the concept of "strategies" by which a formeme or genome may increase its probability of propagating and the "replicate" function that applies a given strategy to a given urban genome. Part III elaborates on and applies such strategies. Finally, "activity costs" refers to the difficulty (whether physical or social) of performing a type of activity in a given space. "Recoding" refers to changes in formemes, and "recoding" cost is the resistance a potential change is likely to meet. These terms give us language to explain how and when U changes (or persists), and to characterize the difference between areas that are geared toward restrictive and highly specific uses/users (those that impose clearly defined costs on specific uses and users) vs. those that are more flexible and less clearly defined. Part II elaborates on these concepts in more detail, and Part III joins them into an integrated model of the sources of urban variants, their differential reproduction, and their degrees of retention.

*5.2. Value of the Model*

The notions of formemes and urban genomes synthesize key strands in the urban studies field. The physical dimension has often been stressed by engineers and planners (see for instance [49,114–116] among others e.g., [21,117]). The fact that areas of the city are geared toward some groups and not others has been a hallmark of urban sociology, which stresses that symbolic boundaries are woven into the fabric of urban space, including some and excluding others, as in the common idiom of "the other side of the tracks" (e.g., [118–121] among others). The nature and distribution of activities across space has tended to be the province of economic geography [55,122]) and urban economists ([123–125]), as well as urban sociologists who see activities as a window into the styles of life the characterize various places (e.g., [126]). Time geographers [57,127] also stress the formation of "activity spaces" [128,129] (that emerge around routines such as work, school, shopping, and recreation.

While these dimensions have often been pursued separately, a sociocultural notion of urban form promises to bring them together and an evolutionary model promises to explain where they come from and how they change. Most crucially, the identification and articulation of a new theoretically defined entity—the formeme—makes it possible to formulate an urban evolutionary model at all. Indeed, without this concept it would be difficult to conceptualize how aspects of urban life vary, survive, and are inherited, at different rates. It is for this reason, we suspect, that prior efforts to develop a formal theory of urban evolution have not been able to move beyond identifying variables in systems dynamics [85] or cellular automata models in which seeds grow to maturity [77]. We call that unit of urban evolution a formeme to convey that it exists to be replicated. This theoretical innovation, however, brings with it a number of advantages, four of which we highlight here: (1) creating the basis for applying the inferential logic of population genetics and evolutionary ecology to urban studies; (2) reversing the analytical lens to the "form's eye view" and the various "survival strategies" that can be formulated from that perspective (evolutionary theory has been an inspiration for this sort of inversion in many domains, but it is not the only one. For example, in his "object-oriented ontology" Harman makes a similar reversal. It would be interesting but beyond the scope of this study to consider the overlaps between this more metaphysical work and the implications of the Darwinian reversal pursued here [130]); (3) encouraging a distributional approach to questions of urban classification (for an account of the advantages of adopting this point of view in evolutionary archaeology, see [131]); and (4) helping us to move from "group" and "developmental" styles of thought toward "tree" or "network" thinking.

First, the formeme concept allows us to extend the basic inferential logic of population genetics and evolutionary ecology into the urban domain. This is because we can think of a "formeme" roughly on the model of a gene or a meme: "urban copy-me" scripts, "urban ideas that spread" or "urban information worth copying". In this case, the information in question consists of information about the physical design of an area, the organization of group life, and the configuration of activities. Formemes can be embedded implicitly in existing urban areas (e.g., the implicit notions about who and what a gated community is for) but they can also be relatively explicitly formulated as building and planning codes that provide templates which, to various degrees of fidelity, determine how urban space is formed. Like bacteria compete for space in bodies and memes compete for space in minds, formemes compete for space in places. As variant formemes are reproduced and disseminated at different rates in different places, urban genomes evolve, producing the continuities and changes of urban life we observe.

Extending the logic of population genetics to formemes moreover makes a seemingly hopeless project of "explaining cities" much more tractable, by redirecting efforts away from fundamentally speculative searches for ultimate origins in the distant past. Consider as an analogy the study of linguistics or religion. Both have over the years been host to intense speculation about origin questions, many answers to which tend to reflect contemporary attitudes projected onto the past (see [24] for a review of the issue in religion). Cities

are no different in this regard, with origin stories ranging from the economic advantages of concentration, defense, or the subjugation of the weak by the powerful Mann ([132] reviews much of the archaeological and anthropological literature). Linguistics, however, to some degree moved beyond speculation and projection by making linguistic features their units: words, grammatical forms, turns of phrase [133]. Basic evolutionary questions then become much more tractable: what accounts for the rise and fall of different patterns of usage? What is it about various linguistic features that make them more or less likely to be retained and circulated? "If you find that a particular concept is very stable in a human group (you can come back later and find it more or less unchanged) it is because it has a particular advantage inside individual minds. If you want to explain cultural trends, this is far more important than tracing the actual historical origin of this or that particular notion. A few pages back, I described the way a Cuna shaman talks to statuettes ... If we want to explain why the Cuna maintain this notion of intelligent statuettes, it does not matter if what happened was that one creative Cuna thought of that a century ago, or that someone had a dream about that, or that someone told a story with intelligent statuettes. What matters is what happened afterward in the many cycles of acquisition, memory and communication" [24] (p. 37). In a similar way, by making formemes our units, we set aside intractable questions about ultimate origins and examine rises and falls in the overall population of formemes. Less important than whatever led some creative person to introduce some new urban variant is what happens afterward, through many cycles of variation, selection, and retention.

Cities in our model are thus various ways of answering the question of "what to be made out of, how it will be arranged, and who and what will it be for". Urban evolution occurs as some answers proliferate and others disappear, and identifying such processes constitutes an explanation of why cities have the global and local shapes and functions they do. These processes can be described formally, and Part III illustrates this direction in more specific detail.

Second, by encouraging us to see the world from the "form's eye" view, the formeme concept helps to reveal how urban evolution can be rational without necessarily being optimized to the interests of any human actors. This is because in many cases urban areas are optimized to the needs of the formeme, namely to reproduce and copy itself. As an illustration, consider an example from Sorensen [87]: urban water systems. In many cities, storm drains are combined with sanitation lines. It is at the very least highly debatable whether this is the most efficient way to manage such systems. What is clear, however, is that from the point of view of a storm drain and sanitation line, their chances of being reproduced increase when packaged together. Combined systems were initially cheaper—in the terminology developed in Part II, the "recoding cost" for retrofitting an area this way was lower than other options, and "even though it is now considered best practice to manage the two water systems separately ... each year that the system grows larger, more households connect to it, more pipes are covered with pavement, and the cost to change to a different system increases" [87] (p. 22). What is best practice from the point of view of contemporary management principles, though, is not necessarily best practice from the point of view of a storm drain "interested" in retaining its place in the urban genome. In other words, the storm drain–sanitation line package represents an evolutionary strategy of co-evolution that increases the survival chances of each, beyond what it would be separately. The result is not necessarily in the contemporary human users' best interest, whose options are constrained by the evolutionary heritage within which they operate.

This perspectival shift therefore opens up to a novel theoretical agenda, dedicated to formulating "survival strategies" by which urban forms may become more or less likely to become adopted and retained by their human hosts, who either remember, institute, and spread them, or leave them to die on the vine. Part III provides an illustrative set of propositions about such strategies, and develops formal language to express them in terms of the differential survival rates of formemes. In more general terms, this shift allows us to take on a key insight from work in organizational diffusion (e.g., [57,134], namely

that organizational types do not necessary spread via rational calculation of managers concerning their optimal efficiency. Other factors include coercion, emulation, and normative authority. Seeing the world from the form's eye point of view, however, allows us to retain the observation about the limits of top-down managerial rationality, without abandoning the proposition that the world embodies some kind of a rational process. Even if that process is not comprehended by the entities that carry it out, it can be modeled and understood. The key question is: rational for what? The formeme concept provides leverage for uncovering the strategies by which urban order emerges as a persistent yet nevertheless temporary settlement among multiple entities—but only once we learn to see ourselves as the competitive environment in which those strategies succeed or fail, and toward which they are directed.

Third, the formeme concept implies a distributional approach to questions about what makes a city the city it is. This is an urban variant of what Ernst Mayr considered one of the core features of the Darwinian breakthrough: "the replacement of essentialism by population thinking" (cf. [106]). The formeme helps to advance this replacement in urban theory, in turn providing a tangible basis for abstractions such as "New York" or "Paris". After all, people do not interact with such abstract entities, let alone the likes of "the post-industrial city", "the creative city", or "the garden suburb". We interact with other individuals, material objects, and the information they communicate. Higher-order structures emerge and evolve as these objects proliferate and combine.

Studying formemes makes this process of the emergent order of cities tangible. Bottom and top are connected via the transmission and replication of formemes, which produce stability and change at the level of populations of urban genomes, that is to say, cities. The neutrality of the formeme in regard to scale allows for propositions about 'bottom up' and planned urban change to coexist in the same framework and ultimately in the same empirical studies. It also facilitates the study of how infrastructures and enterprises and activating ideas propagate simultaneously at multiple scales. As we will see in Part III, the spatial diffusion of forms is awkwardly studied by incumbent approaches be they single place case studies or highly-powered spatial statistics studies.

By grounding the "character of a city" in its population and distribution of formemes, the formeme concept therefore gives concrete meaning to seemingly ethereal ideas like "urban culture". In this view, as in anthropological applications of meme/diffusion theory (cf. [24]), "culture is a name for similarity". A culture is not for example some essence that some places share, such as "Americans value individual achievement" or "Asian cultures value the family"—any more than a gene for brown eyes is some entity that brown-eyed people share bits of. Rather, some cities have similar distributions of formemes, and in virtue of this, are more geared toward similar notions of what groups and activities are supposed to occur there. Where there is more, say, family-oriented parks, restaurants, schools, we can speak of cities or areas with similar cultures. This in turn makes the task of classification—urban phylogenetics—into an empirically and theoretically tractable task of creating concepts, tools, and databases for identifying varying degrees of overlap in formemes across space and time.

Similarity is not the same as sameness, and the distributional approach implied by the formeme concept has the additional advantage of facilitating questions about degrees of conflict or overlap across scales within cities. Again the comparison to culture is informative. Not everybody wholeheartedly accepts a given cultural norm, and many of us live with varying degrees of conflict or ambivalence [135]. This situation is not intelligible if we think of "culture" as a unified entity that stands outside of us, which we take on as a totality. The distributional approach by contrast means we can identify and measure how broadly a single urban genome permeates all areas of a city, as well as the degree to which disagreement or diversity exist within and across various locations. This approach provides a powerful way to pursue in precise terms questions about the degree of conflict, diversity, and agreement in urban life.

Fourth, the formeme concept helps us to formulate evolutionary models in the mode of "tree" or "network" thinking in contrast to "group" or "developmental" models [136]. Group thinking considers each member of a group as an independent replicate. Each instance tells us something about the class as such, and the analytical goal is to abstract from the instances to understand causes of the generic group. Evolving entities, however, are not independent replicates: each instance inherits some features from its predecessors, and passes others on. A given forememe is part of an interconnected set of formemes, in which its elements share features derived from ancestors. Whether this set forms a tree or a web is an open empirical question. In any case, if we are interested in why a set of ten urban forms exhibit a particular trait, we will need to ask if this represents ten separate originations of that trait, eight with two subsequent differentiations, or four, or perhaps one, independent origination, with all ten inheriting the same formeme [136]. Similarly, whether as webs or trees, the evolution of formemes involves complex branchings and combinations rather than a linear development. The evolutionary question is not which city or form is farthest along a universal path of development, but rather where a given entity sits within the broader "tree of urban life".

In sum, our model emerges from out of both the transdisciplinary urban studies literature and the general sociocultural literature. From the former, we compile key but partial insights that we seek to synthesize into a more complete theory of urban evolution. From the latter we take the general elements that a model of urban evolution should possess. Our model aims to put the two together. In this model, formemes differentially survive and reproduce by way of their capacity to recruit human attention, interest, and interaction: variant types of industrial districts, food production and delivery networks, education and social service provision systems, housing concepts, zoning concepts and building codes, road patterns, park designs, community organizations, and more spread as some variants prove more viable in some niches than others. Accordingly, their fate is crucially bound up with the varying capacities, resources, tastes, needs, and values of human actors.

## 6. Discussion and Conclusions of Part I

By way of conclusion, it is worth reviewing some of what the approach to urban evolution we have articulated does and does not imply.

1.  A model of sociocultural urban evolution does not require telling the story of the evolution of cities from their initial appearance some 9500 years ago. Darwin studiously avoided speculations about the origins of life and instead developed a theory of evolutionary mechanisms and processes. Similarly, we believe it is more productive at present to place speculations about the origins of urban life to one side and focus our efforts on generic evolutionary processes. That said, we view urban evolution as a breakthrough event in human history. If cultural evolution permits information to be stored and transmitted outside of genetics in books, language, and the like, rapidly increasing the pace of evolution, urban evolution transforms the physical contexts in which cultural information exists.

2.  A model of sociocultural urban evolution is neither reductionist nor deterministic. It is not reductionist since it makes no claim that sociocultural processes are reducible to genetic imperatives, though it is open to dynamic gene–culture interactions. It is not deterministic in that its basic processes are probabilistic, involve complex interactions, and multiple overlapping principles of selection, survival, retention, and replication, not to mention co-evolution.

3.  A model of sociocultural urban evolution is not necessarily progressivist or teleological. Many urban forms we might deem pernicious may also become fruitful and multiply. While there may be a generic tendency toward greater levels of complexity and differentiation, this is neither necessary nor necessarily good. Divergence is not the same as development.

4.  A model of sociocultural urban evolution is not necessarily concerned with grand narratives or world-historical developments. Darwin built his theory (in part) by

observing the distribution of characteristics among the humble finches in the Galapagos. Similarly, we may develop generic mechanisms of urban evolution through observations of mundane features of urban life, such as cul-de-sacs, pizza restaurants, block patterns, bohemian neighborhoods, plazas, porches, or public art, and build out from there.

5. A model of sociocultural urban evolution does not imply a slavish mimicry of biological principles of evolution. There is no expectation to find one-to-one correspondences between biological and urban evolution. One may provide inspiration for the other, they both might be instances of a general type of theory, but each involves a host of domain-specific mechanisms.

6. A model of urban evolution is neither top-down nor bottom-up, neither pro- nor anti-planning. While "order without design" at times has become a slogan for evolutionist thinking in the social sciences, this need not imply any antipathy toward formal urban planning or the effort to intelligently guide urban development. In fact, we may view formal planning as an emergent activity that accelerates the codification of successful formemes. It does, however, suggest that formal plans are only one input into highly complex interactions (including the plans of numerous individuals and groups), where the relationship between intention and outcome is tenuous and subject to surprise and contingency.

7. An evolutionary approach suggests embracing new metaphors for the role of the planner: the planner less as an engineer pulling the levers of a well-tuned machine and more as a gardener in a forest, seeking to cultivate a rich ecosystem while remaining sensitive to processes unfolding through their own dynamics. Scheer [98] elaborates on the planning and design implications of this evolutionary perspective in greater detail. On one end, it suggests caution in imposing visionary planning ideas onto an ecology that has evolved as a complex of solutions to real problems. On the other, it views reality as itself an ongoing experimentation from which new solutions may be drawn and attempted in new contexts. What is there constrains what is possible, but multiple possibilities are available in what is there. The creativity of planning comes from recognizing these and contributing to the unfolding experimentation already occurring. In addition, as Sheer notes, this perspective suggests paying attention to varying rates of change: deeply-rooted patterns such as lots and blocks are difficult to change and tend to recur, and any new buildings or objects (such as signs or fences or trees) will be shaped by those slower-moving elements; smaller changes to groups and activities may have more limited changes to the physical forms. Our model provides tools for formulating these and other propositions about the impact of planning, developed further in Part III; see also [137].

Parts II and III build upon the ground laid in Part I. Part II formally defines the central terms of the model, and articulates core functions based on these definitions. Part III elaborates on the model in terms of evolutionary processes: sources of variations, mechanisms of selection, means of retention. A concluding discussion takes stock of the effort, and discusses limitations, future directions, and challenges. Part IV applies our model to Yelp from the cities of Toronto and Montreal, demonstrating how formemes can be used to measure evolutionary distance. While the world will no doubt continue to change, often in unexpected and potentially cataclysmic ways that demand attention and research, the main goal of this research is on another level: not to keep up with the latest news but to elaborate on how to make and understand this news in new ways.

**Author Contributions:** Conceptualization (D.S., M.S.F. and P.A.); writing, all phases (D.S., M.S.F. and P.A.). All authors have read and agreed to the published version of the manuscript.

**Funding:** This work was supported, in part, by a University of Toronto Connaught Global Challenge Award, and the School of Cities Urban Challenge Fund.

**Acknowledgments:** We thank for their input during the development of this work Rob Wright, Ultan Byrne, Khalil Martin, Noga Keidar, Fernando Calderon Figueroa, Clara Bitter, Andre Sorenson, Marion Blute, Juste Raimbault, Yaara Rosner-Manor, Fabio Dias, and Abid Mehmood.

**Conflicts of Interest:** The authors declare no conflict of interest.

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
