# Peer review of "Towards a Model of Urban Evolution—Part I: Context"

_urbansci, doi:10.3390/urbansci6040087_

Round 1

Reviewer 1 Report

I know the evolutionary literature better than the urban literature but it seems to me that this paper has done an excellent job in beginning to relate them. A couple of heads up. I have no problem with the neologism "formeme" but most sociocultural evolutionists these days do not like or use the meme meme (for no very good reason actually) which might have a small negative effect on the reception of this work among them. Secondly, as the paper notes, there are several formulations of a basic evolutionary model but for the one the paper puts up front, "variation, selection and retention", you cite Aldrich which is o.k. but it was originally Campbell's. I do note like it much because it seems to emphasize survival over transmissibility but that is clearly not how it is used throughout the paper so I guess I wouldn't worry about it.

Author Response

Reviewer Comment: “I know the evolutionary literature better than the urban literature but it seems to me that this paper has done an excellent job in beginning to relate them. A couple of heads up. I have no problem with the neologism "formeme" but most sociocultural evolutionists these days do not like or use the meme meme (for no very good reason actually) which might have a small negative effect on the reception of this work among them. Secondly, as the paper notes, there are several formulations of a basic evolutionary model but for the one the paper puts up front, "variation, selection and retention", you cite Aldrich which is o.k. but it was originally Campbell's. I do note like it much because it seems to emphasize survival over transmissibility but that is clearly not how it is used throughout the paper so I guess I wouldn't worry about it.”

Response: Thank you for this comment. We agree that Campbell should be referenced, and have added references to his key articles on pp 2 and pp 11. We also added a footnote about transmissibility on p. 12. You could be right regarding the “meme meme,” but that is a risk we’ll take.

Reviewer 2 Report

In the introduction, the study’s contribution to the literature could be clarified.

Author Response

Reviewer Comment: “In the introduction, the study’s contribution to the literature could be clarified."

Response: Thank you for asking for this clarification. On p. 4, we add added this sentence: “This synthesis of various strands of urban theories with sociocultural theories of evolution is the primary contribution of this paper.”

While the above sentence captures the main contribution of this specific paper, we note that it is also the first in a series of papers, and in that context it provides context and background to the overall project of developing a model of urban evolution.

Reviewer 3 Report

Dear Authors, please find some comments/feedback in the brief list below:

1. You clearly ground your whole research on stretching/applying Darwin's theory of evolution to the "urban"... but this raises one fundamental questions: what is "urban"? "Urban" is not synonymous with "city" [a great number of scholars have agreed on that] and furthermore, city is not an unambiguously recognized, let alone describable, category [a favela in Brazil, a slum in India, a European metropolis or a historic village, an American town, are not born, or develop, or potentially evolve in similar ways, and to understand this complexity within a single model is in my opinion unrealistic and also conceptually wrong]. Thus, I would suggest an initial definition of what you, in your study, assume as "urban" and as "city." Moreover, in Part IV you concentrate on the case study of Toronto -which is not any city, even though you stress a few times the intention of your model to overcome site specificities-, so I guess you should be clear right from the start.

2. on the definition of complexity also applied to "urban living" and "city dynamics" I would suggest you to also refer to the work of Edgar Morin.

3. besides the biological metaphor [greatly applied to cities by many important theorists over the past centuries, as well as it has been opposed by many others] there are other, more recent theories trying to shift away from the "anthropocentric view"; the first that comes to my mind, applied to urban studies, is the object-oriented ontology (Harman, and then Morton) which could be interesting for you to investigate a bit or integrate in your discussion.

4. "Discussion and Conclusion of Part I" appears to be more of a recap of some already stated concepts than a proper discussion... why did you develop this "model of urban evolution" and to what specific purpose? Who will benefit from it? how can it be applied to city planning/management/studies...? Obviously, not seeing the model at the end of your discussion does not help the overall understanding of it.

5. I saw online that this very contribution was already issued on October 24, 2020; despite a global epidemic, a war, an energetic crisis and so forth, nothing has changed in your research? I believe it would be very interesting to understand if and why so.

Author Response

Reviewer Comment 1:  "You clearly ground your whole research on stretching/applying Darwin's theory of evolution to the "urban"... but this raises one fundamental questions: what is "urban"? "Urban" is notsynonymous with "city" [a great number of scholars have agreed on that] and furthermore, city is not an unambiguously recognized, let alone describable, category [a favela in Brazil, a slum in India, a European metropolis or a historic village, an American town, are not born, or develop, or potentially evolve in similar ways, and to understand this complexity within a single model is in my opinion unrealistic and also conceptually wrong]. Thus,I would suggest an initial definition of what you, in your study, assume as "urban" and as "city." Moreover, in Part IVyou concentrate on the case study of Toronto -which is notanycity, even though you stress a few times the intention of your model to overcome site specificities-, so I guess you should be clear right from the start.”

Response 1: Thank you for this comment. This is clearly a large and difficult philosphical issue. In the present context, we believe the most appropriate thing for us to do is to indicate our general stance on the question. To that end, we added the following footnote top. 14: “In aspiration, our approach is similar to that of Scott and Storper (2015, 2016) in seeking general processes that hold across time and space. This is in keeping with the promise of Darwinian theory in general, to bring the diversity of natural evolution under a common model. Even so, such an approach requires close examination of local and special mechanisms by which the generic process of variation-selection-retention occurs. Nevertheless, in contrast to urban theorists like Scott and Storper, we are not developiong a theory of urbanization per se but rather elaborate the terms that make formulating hypotheses about urbanization and other aspects of urban evolution possible. For this reason in the present work we are agnostic regarding the definition of concepts like “city,” since we view that too as a result of evolutionary processes to be investigated.”In addition and as a guide to readers who might otherwise be confused, we have defined urbanization/urban on the bottom of page 1. According to this definition, urban and “city” are nearly synonomous.

Reviewer Comment 2: "on the definition ofcomplexityalso applied to "urban living" and "city dynamics" I would suggest you to also refer to the work of Edgar Morin.”

Response 2: Thank you for alerting us to Morin’s work on complexity, who takes a more epistemological approach. We note this alternative in a footnote on p. 7:

“Elsewhere, philosophers like Edgar Morin have discussed complexity as a core epistemological concept and a successor to rationalistic modes of thinking. Our framework is more applicable to theauthors we have cited who are interested in dynamic models of social ssystems such as cities.”

Reviewer Comment 3: "besides the biological metaphor [greatly applied to cities by many important theorists over the past centuries, as well as it has been opposed by many others] there are other, more recent theories trying to shift away from the "anthropocentric view"; the first that comes to my mind, applied to urban studies, is the object-oriented ontology (Harman, and then Morton) which could be interesting for you to investigate a bit or integrate in your discussion.”

Response 3: Thank you for this comment. We agree that there are other philsophical traditions that seek to shift from the anthropecentric view, such as Harman. We acknowldge this in this footnote to p. 17:

“Evolutionary theory has been an inspiration for this sort of inversion in many domains, but it is not the only one. For example, “object-oriented ontology” (Harman 2018) makes a similar reversal. It would be interesting but beyond the scope of this study to consider the overlaps between this more metaphysical work and the implications of the Darwinian reversal pursued here.”

Reviewer Comment 4: ""Discussion and Conclusion of Part I" appears to be more of a recap of some already stated concepts than a proper discussion... why did you develop this "model of urban evolution" and to what specific purpose? Who will benefit from it? how can it be applied to city planning/management/studies...? Obviously, not seeing the model at the end of your discussion does not help the overall understanding of it."

Response 4: We have added some additional comments that connect this study to some possible benefits for planning studies, though we re-iterate that our primary goal is the theoretical synthesis referenced above.

“An evolutionary approach suggests embracing new metaphors for the role of the planner: the planner less as an engineer pulling the levers of a well-tuned machine and more as a gardener in a forest, seeking to cultivate a rich ecosystem while remaining sensitive to processes unfolding through their own dynamics. Sheer (2017) elaborates the planning and design implications of this evolutionary perspective in greater detail. On one end, it suggests caution in imposing visionary planning ideas onto an ecology that has evolved as a complex of solutions to real problems. On the other, it views reality as itself an ongoing experimentation from which new solutions may be drawn and attempted in new contexts. What is there constrains what is possible, but multiple possibilities are available in what is there. The creativity of planning comes from recognizing these and contributing to the unfolding experimentation already occurring. In addition, as Sheer notes, this perspective suggests paying attention to varying rates of change: deeply-rooted patterns such as lots and blocks are difficult to change and tend to recur, and any new buildings or objects (such as signs or fences or trees) will be shaped by those slower-moving elements; smaller changes to groups and activities may have more limited changes to the physical forms. Our model provides tools for formulating these and other propositions about the impact of planning, developed further in Part III (see also Calderon-Figuerora, Silver, and Bidian (2022).”

Reviewer Comment 5: " I saw online that this very contribution was already issued on October 24, 2020; despite a global epidemic, a war, an energetic crisis and so forth, nothing has changed in your research? I believe it would be very interesting to understand if and why so.”

Response 5: Thank you for noting this.While we like everybody else have been affected by the major events of recent years, our goal in this work is to formulate theory at a general level. We have of course been continuing to extend our model into new domains and applications, but that shows up in other papers.

To capture this spirit we have added the following sentence to p. 21: “While the world will now doubt continue to change, often in unexpected and potentially cataclysmic ways that demand attention and research, the main goal of this research is on another level: not to keep up with the latest news but to elaborate how to make and understand news in new ways.”

Reviewer 4 Report

The paper presented turns out to be excellent in its current form.

We obviously look forward to subsequent papers to learn about the evolution of the work on the structured urban evolutionary model.

More attention should be paid to the editorial standards of the journal, such as footnotes and bibliographical references.

Author Response

Reviewer Comment: "More attention should be paid to the editorial standards of the journal, such as footnotes and bibliographical references."

Response: The manuscript has been modified to conform to editorial standards.